# Update and New Epidemiological Aspects about Grapevine Yellows in Chile

**DOI:** 10.3390/pathogens9110933

**Published:** 2020-11-11

**Authors:** Nicolás Quiroga, Camila Gamboa, Daniela Soto, Ana Maria Pino, Alan Zamorano, Juan Campodonico, Alberto Alma, Assunta Bertaccini, Nicola Fiore

**Affiliations:** 1Departamento de Sanidad Vegetal, Facultad de Ciencias Agronómicas, Universidad de Chile, La Pintana, 8820808 Santiago, Chile; nicolasquirogabarrera@gmail.com (N.Q.); camila.gamboa@ug.uchile.cl (C.G.); soto.sandoval.d@gmail.com (D.S.); anamariapinoelgueta@gmail.com (A.M.P.); agezac@u.uchile.cl (A.Z.); 2Programa de Doctorado en Ciencias Silvoagropecuarias y Veterinarias, Campus Sur Universidad de Chile. Santa Rosa 11315, La Pintana, 8820808 Santiago, Chile; 3Ph.D. Program in Science, Ecology and Evolution mention, Facultad de Ciencias, Universidad Austral de Chile, 5091000 Valdivia, Chile; juan.campodonico@ug.uchile.cl; 4Forest and Food Sciences and Technologies, Department of Agricultural, Università di Torino, DISAFA, 10095 Grugliasco (TO), Italy; alberto.alma@unito.it; 5Alma Mater Studiorum, Department of Agricultural and Food Sciences, Università di Bologna, 40127 Bologna, Italy; assunta.bertaccini@unibo.it

**Keywords:** insects vector, nested-PCR/RFLP, sequencing, weeds

## Abstract

To date, phytoplasmas belonging to six ribosomal subgroups have been detected to infect grapevines in Chile in 36 percent of the sampled plants. A new survey on the presence of grapevine yellows was carried out from 2016 to 2020, and 330 grapevine plants from the most important wine regions of the country were sampled and analyzed by nested PCR/RFLP analyses. Phytoplasmas enclosed in subgroups 16SrIII-J and 16SrVII-A were identified with infection rates of 17% and 2%, respectively. The vineyards in which the phytoplasma-infected plants were detected were further inspected to identify alternative host plants and insects of potential epidemiological relevance. Five previously unreported plant species resulted positive for 16SrIII-J phytoplasma (*Rosa* spp., *Brassica rapa*, *Erodium* spp., *Malva* spp. and *Rubus ulmifolius*) and five insect species were fully or partially identified (*Amplicephalus ornatus*, *A. pallidus*, *A. curtulus*, *Bergallia* sp., *Exitianus obscurinervis*) as potential vectors of 16SrIII-J phytoplasmas. The 16SrVII-A phytoplasmas were not detected in non-grape plant species nor in insects. This work establishes updated guidelines for the study, management, and prevention of grapevine yellows in Chile, and in other grapevine growing regions of South America.

## 1. Introduction

Viticulture is one of the most important agricultural activities of Chile, where the export of wine and the socio-cultural component are its main characteristics. The total area of vineyards for winemaking is approximately 141,000 hectares. Chile is currently the leading wine exporter of the American continent and the fourth world exporter, with a total of 966.2 million liters exported for a value of US $2066.1 in 2018 (Pizarro, 2019) [1]. 

The presence of phytoplasmas (grapevine yellows, GY) in vineyards in Chile has been reported since the late 1980s; however, the first molecular studies on their identity and distribution began 20 years later [2,3,4]. These phytoplasmas resulted in being enclosed in six ribosomal groups. The first survey for the identification of phytoplasma in grapevine was carried out from 2002 to 2006 in vineyards located in the regions of Coquimbo, Valparaíso, Libertador General Bernardo O’Higgins, Maule, and Metropolitana, where the main wine production is concentrated. Mixed and single phytoplasma infections were detected in 34 out of the 94 samples collected (36%). The identified phytoplasmas and their prevalence was: 16SrI-B (7.4%), 16SrI-C (‘*Candidatus* Phytoplasma asteris’) (4.3%), 16SrVII-A (‘*Ca.* P. fraxini’) (13.8%), and 16SrXII-A (‘*Ca.* P. solani’) (16.0%). In 2010, the phytoplasmas 16SrIII-J (‘*Ca.* P. pruni’-related) and 16SrV-A (‘*Ca.* P. ulmi’) were found infecting two vineyards of cultivars Superior and Syrah, respectively. Out of the six plants analyzed, three with symptoms and three without the presence of phytoplasma was detected only in those symptomatic. It is important to note that the “flavescence dorée” (FD) phytoplasma and its insect vector, the leafhopper *Scaphoideus titanus* Ball, have never been found in Chile [5,6,7,8]. Epidemiological studies on GY have verified the presence of three phytoplasma reservoir weeds species, *Convolvulus arvensis* L., *Galega officinalis* L., and *Polygonum aviculare* L. [9,10]. Moreover, two insect species, *Paratanus exitiosus* (Beamer) and *Bergallia valdiviana* Berg, have been described as vectors of the 16SrIII-J phytoplasma [11]. These insects feed on weeds and only occasionally on grapevine, allowing the transmission of phytoplasmas to this species. In Chile, there are no planthoppers and leafhoppers, which cause direct economic damage to the grapevine. 

Since grapevine is economically important for Chile and due to the repeated detection of phytoplasmas in the crop, the present work aimed to provide further epidemiological information about phytoplasmas infecting this plant species in the country. For this purpose, in the new survey conducted, the presence of undescribed potential insect vectors as well as new reservoir plant species in infected vineyards is reported.

## 2. Results

### 2.1. Phytoplasma Detection in Grapevine and Other Plant Species in the Vineyard 

From the end of January and throughout the autumn, phytoplasma infected grapevine plants showed characteristic GY symptoms that were independent from the phytoplasma identity. The main symptoms were downward rolling and reddening of the leaves and leaf veins (red cultivars) or yellowing of the leaves and leaf veins (white cultivars); drying up of bunches; and incomplete shoot lignification (Figure 1 and Figure 2). Sixty-seven vineyards were sampled: 15 in the Valparaíso region, 14 in O’Higgins, and 38 in Maule, with eight cultivars (Cabernet Sauvignon with 72 samples, País 55, Chardonnay 52, Carménère 51, Sauvignon blanc 41, Pinot noir 29, Tintorera 23, and Semillón with seven). The sampling of shoots and leaves was performed from 330 grapevine plants and 80 different plant species (weed, woody, and shrub) from 2016 to 2020. Amplicons of R16F2n/R2 (1250 bp) from 16S rRNA gene and of ItSSu12pFn/ItSSu12pR2n (820 bp) that amplify the full length of the *SSu12*p gene and the partial sequence of the *SSu7*p gene were obtained by PCR and nested PCR from 63 grapevine and 19 non-grape plant species collected in the same vineyards. Consensus sequences obtained from clones were used to perform the phylogenetic and restriction polymorphism analyses. Phytoplasmas belonging to the 16SrIII-J (17%) and 16SrVII-A (2%) ribosomal subgroups were identified in 55 and eight grapevine plants, respectively (Table 1). The 16SrVII-A phytoplasmas were detected only in grapevine plants of the symptomatic Cabernet Sauvignon cultivar (Figure 2). In the other plant species, the 16SrIII-J phytoplasma was detected (Table 2). The symptomatic weeds and bushes that were positive for the 16SrIII-J phytoplasma were *Rosa* spp. (Family *Rosaceae*—seven samples), *Brassica rapa* L. (Fam. *Brassicaceae*—four samples), *Malva* spp. (Fam. *Malvaceae*—two samples), *Erodium* spp. (Fam. *Geraniaceae*—three samples), and *Rubus ulmifolius* Schott (Fam. *Rosaceae*—three samples) (Table 2). In *Rosa* spp., witches’ broom, leaf deformation, phyllody, and virescence were observed. In *B. rapa* and *Malva* spp., corky texture of the leaves and deformation were also detected. In *Erodium* spp., redness of the leaves and dwarfism, and finally, in *R. ulmifolius,* a decrease in the size of the leaves and witches’ broom were observed (Figure 3).

### 2.2. Phytoplasma Detection in Insects

Insects were captured from three phytoplasma positive vineyards of cultivar Pinot noir in the locality of Casablanca (Valparaiso region), Carménère in San Javier (Maule region) and Cabernet Sauvignon in Marchigue (O’Higgins region). The analyzed insects totaled 1126. The 16SrIII-J phytoplasma was detected in seven leafhopper species: *P. exitiosus*, *B. valdiviana*, *Bergallia* sp., *Amplicephalus ornatus* Linnavuori, *Amplicephalus pallidus* Linnavuori, *Amplicephalus curtulus* Linnavuori & DeLong, and *Exitianus obscurinervis* Stål (Figure 4). The 16SrVII-A phytoplasma was not detected in any of the insects (Table 2).

### 2.3. Molecular Characterization of Phytoplasma Strains

Sequences of the 16SrIII-J phytoplasma were 100% identical at the nucleotide level. These sequences were identical to the one of the strain Vc33 from periwinkle in Chile (GenBank accession number LLK01000000). The 16SrIII-J sequences formed a monophyletic group, as shown by neighbor joining analyses (Figure 5). In addition, the sequence of the single grapevine plant infected with 16SVII-A phytoplasma shared 99.9% nucleotide identity with the sequence of the ‘*Ca*. P. fraxini’ from ash yellows, strain ASHY4 reported in EPPO-QBank (strain number QPh42) for the 16S rRNA gene, and 100% of identity for the *SSu12p* gene amplicon sequence obtained from an ASHY strain infecting periwinkle from the EPPO-QBank collection maintained at the University of Bologna (GenBank accession number MT161529). The identification of the phytoplasmas found in plants and insects was confirmed through in silico restriction fragment length polymorphism (RFLP) analyses (Figure 6 and Figure 7).

## 3. Discussion

During this survey, phytoplasmas enclosed in the 16SrIII-J and 16SrVII-A subgroups were detected in vineyards of Chile. Five wild plant species were positive to phytoplasma 16SrIII-J (*Rosa* spp., *B. rapa*, *Erodium* spp., *Malva* spp., and *R. ulmifolius*) as well as five insects (*A. ornatus*, *A. pallidus*, *Bergallia* sp., *A. curtulus,* and *E. obscurinervis*). These results associate the GY presence exclusively with two phytoplasmas of which 16SrIII-J was prevalent; this result differed from those reported in previous surveys carried out more than 10 years ago [4,5,6]. This finding is very likely related to the wide presence of *P. exitiosus* and *B. valdiviana* in the country, and to the wide presence of plant species other than grapevine that harbor the pathogen.

The 16SrIII-J phytoplasma is mainly reported in South America and Mexico [12,13,14,15,16,17,18]. In Chile, historically, this phytoplasma has been detected in sugar beet [19]; however, it is now described as infecting a wide range of herbaceous and woody plant species [10,20,21,22,23]. This phytoplasma is transmitted by two leafhopper species *P. exitiosus* and *B. valdiviana* [11].

In this study, the epidemiological survey focused mainly on the Maule region (Maule valley), which has not been previously monitored. In these vineyards, an apparent high activity of *P. exitiosus* was observed. This was the most abundant insect captured, and 160 out of 220 individuals collected were positive for the 16SrIII-J phytoplasma. A special situation was observed in the red cultivar País, where plants infected by phytoplasma 16SrIII-J only showed downward rolling and yellowing of the leaves, but not reddening of leaves, behaving like the symptoms observed on white cultivars. This could be related to the fact that the cultivar was introduced to Chile in the 16th century, adapting to the edaphoclimatic conditions of the country, thus developing high vigor and tolerance to biotic and abiotic stresses.

In the other regions surveyed, a greater diversity of insect vectors and potential vectors was observed and transmission tests are ongoing to determine the transmission ability of *A. curtulus, A. pallidus*, *A. ornatus*, *Bergallia* sp., and *E. obscurinervis*.

This is the first report of 16SrIII-J phytoplasma infecting *Rosa* spp. and *R. ulmifolius* in Chile. *Rosa* spp. is used in vineyards as an indicator plant for the foliar fungus *Uncinula necator*. In addition, it is traditionally planted to decorate the vineyards in central Chile. *R. ulmifolius* is used as a living fence and sometimes grows spontaneously near the vineyards.

Based on these results, it seems that new species of Auchenorrhyncha could be involved in the transmission of phytoplasmas in the Chilean vineyards. Furthermore, *B. campestris*, *Malva* spp., and *Erodium* spp. are weeds that begin their life cycle after the first autumn rain and stay green until the end of the spring. This means that these weeds may play a relevant role as a reservoir of the 16SrIII-J phytoplasma during the grapevine winter recess, representing a constant source of inoculum.

Several weeds have been described as reservoirs of the 16SrIII-J phytoplasma in Chile such as *C. arvensis*, *P. aviculare*, and *G. officinalis*; these are generally asymptomatic, sometimes showing, however, small yellow leaves [9,21].

The phytoplasma 16SrVII-A has been previously detected in grapevines, weeds, and shrub species from southern Chile [*Gaultheria phillyreifolia* (Pers.) Sleumer and *Ugni molinae* Turcz.] and peony *(Paeonia lactiflora* Pall.). An insect vector for this phytoplasma has not been identified; however, three species of leafhoppers have been reported as potential vectors: *P. exitiosus*, *Carelmapu ramosi* Linnavuori, and *Carelmapu aureonitens* Linnavuori & DeLong [5,6,10,24,25,26]. The drying up of bunches observed in the cv. Cabernet Sauvignon was associated with the presence of 16SrVII-A phytoplasma (Figure 2B), with approximately 6% of product losses.

GY in Chile appears to now be mainly associated with the 16SrIII-J and 16SrVII-A phytoplasmas. It should be emphasized that the infected insect species, vector, and potential vector feed only occasionally in grapevine plants, mainly in late summer when the weeds finish their cycle.

The results of this new survey confirmed the absence of the phytoplasma associated with FD in Chile, and of its insect vector *S. titanus*. However, the situation must be constantly monitored considering the results of climatic studies indicating that the insect vector would have the optimal conditions to establish in Chile after a possible accidental introduction [27].

## 4. Materials and Methods 

### 4.1. Grapevine Plants Survey

From September 2016 to April 2019, symptomatic and asymptomatic grapevine samples were collected from the most important wine producing areas of Chile, located in the Valparaíso, Libertador Bernardo O’Higgins, and Maule regions. Each sample corresponded to four 30 cm-long shoots with at least five leaves, collected from a single plant. The samples were transported in thermo regulated containers and stored at 4 °C prior to the extraction of nucleic acids.

### 4.2. Plants Other than Grapevine and Insect Sampling 

From September 2017 to August 2018, three 16SrIII-J infected vineyards were visited every 15 days to capture insects. The vineyard planted with cv. Pinot Noir was located in Casablanca (Valparaiso region), Carménère in San Javier (Maule region), and Cabernet Sauvignon in Marchigüe (O’Higgins region). The insect trapping was carried out through an entomological sweeping net doing 200 sweeping in 20 linear meters. directed at the weeds present in the vineyard. The captured individuals were placed in plastic tubes sealed with a cotton ball; a tiny piece of weed was added inside to avoid dehydration until identification. The insects were moved to the laboratory in a 15 L icebox at 15 °C, separated by morphological characteristics and analyzed in groups of five individuals. Male insects in 70% ethanol were sent to the University of Turin, Italy, and Universidad Austral to be classified based on their genitalia, which were examined using a stereo-microscope. The identification of Auchenorrhyncha and Sternorrhyncha was then obtained primarily using the dichotomous keys of Ribaut (1936, 1952) [28,29]. The nomenclature was reviewed mainly on the basis of della Giustina (1989) [30]. Weeds, shrub, and non-grapevine woody plants from different botanical families present in the vineyards and in their surroundings were randomly collected. 

### 4.3. Phytoplasma Detection

Total nucleic acids were extracted from leaf midribs and insects using a CTAB method [31], dissolved in Tris-EDTA pH 8.0 buffer and maintained at 4 °C. PCR amplification was carried out using 20 ng μL^−1^ of nucleic acid. Direct and nested PCR assays on the 30 S ribosomal protein *SSu12p* gene were performed according to the protocol of Cui et al. [32]. Further PCR with primer pair P1/P7 [33,34] and nested PCR with R16F2n/R2 primers amplifying the 16S rRNA gene [35] were performed as described by Schaff et al. [36]. Nested PCR amplicons from both genes were purified using the EZNA Gel Extraction Kit (OMEGA Bio-tek, Norcross, GA, USA). DNA fragments were ligated into the pGEMT-Easy Cloning Kit (Promega, Madison, Wisconsin, USA). Putative recombinant clones were analyzed by colony PCR using at least five colonies, and selected fragments from three colonies per sample were sequenced in both directions at Macrogen USA Corp. The sequences were then aligned with those of identified strains deposited in GenBank using the BLAST engine for local alignment (version BLAST N2.2.12). Phylogenetic trees were constructed in MEGA software v7.0.26 using the neighbor-joining algorithm and maximum composite likelihood method to estimate distances with 1000 bootstrap replicates. *Acholeplasma laidlawii* was employed as an out-group to root the tree.

The phytoplasma identification was carried out using in silico RFLP analysis with *Hha*I, *BstU*I *Alu*I, and *Taq*I restriction enzymes in the *i*PhyClassifier online tool (https://plantpathology.ba.ars.usda.gov/cgi-bin/resource/iphyclassifier.cgi), which was also used to calculate the sequence similarity scores [37]. 

## 5. Conclusions

16SrIII-J is the main phytoplasma detected in the last five years in the GY infected vineyards in Chile. Two GY insect vector species have been described and through this study, other potentially vector species were found. In addition, new shrub species and reservoir weeds of this phytoplasma were described. This update will allow establishing new guidelines for the study, management, and prevention of GY disease in Chile, and in other grapevine growing regions of South America, mainly Argentina and Peru, that share with Chile an important increase in viticulture and several species of spontaneous flora and insect fauna. Transmission trials are in progress to determine if the Auchenorrhyncha reported as harboring are also vectoring the 16SrIII-J phytoplasma.

## Figures and Tables

**Figure 1 pathogens-09-00933-f001:**
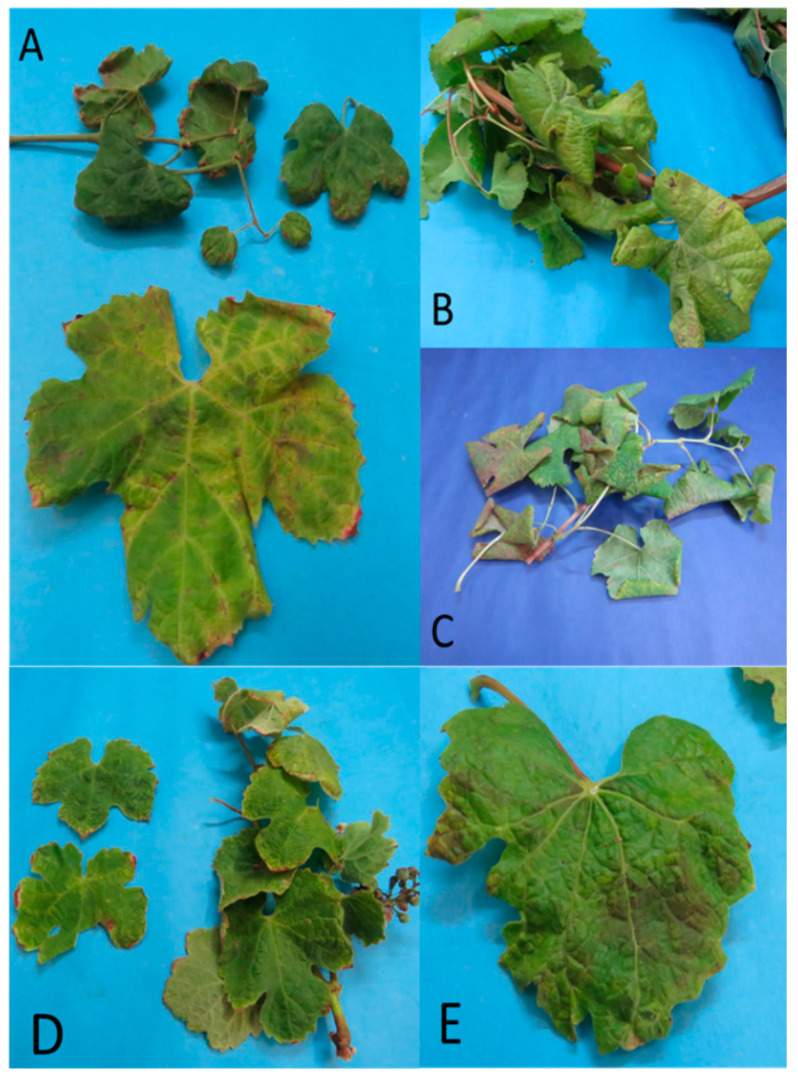
Grapevine leaves showing symptoms associated with the presence of 16SrIII-J phytoplasmas. (**A**) VN17, cultivar Chardonnay with leaf deformation and yellow vein; (**B**) VN12 cultivar País with downward rolling of leaves and generalized yellowing; (**C**) VN29 cultivar Tintorera with downward rolling of leaves and generalized yellowing; (**D**) VN69 cultivar Semillón with small leaves; and (**E**) VN32 cultivar Sauvignon blanc with deformation and mild downward rolling of the leaf.

**Figure 2 pathogens-09-00933-f002:**
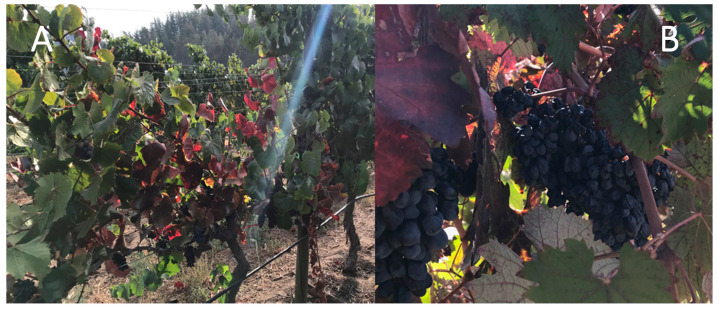
Cabernet Sauvignon symptoms associated to 16SrVII-A phytoplasma presence. (**A**) VN137 with reddening of leaves and leaf veins at the end of summer; (**B**) VN315 with reddening of leaves and leaf veins, and drying up of bunches at the end of summer.

**Figure 3 pathogens-09-00933-f003:**
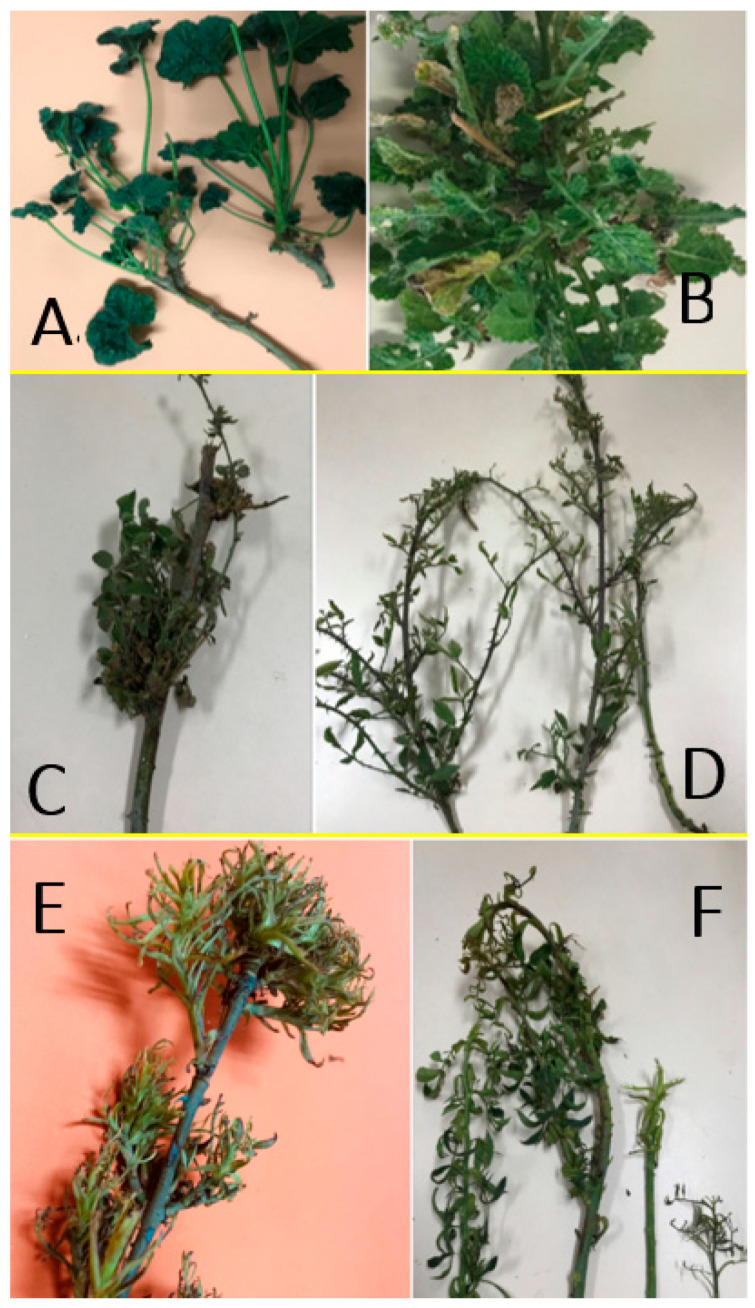
Symptoms in weeds and shrubs infected with 16SrIII-J phytoplasma. (**A**) Corky leaves in *Malva* spp.; (**B**) deformation and corky texture of leaves in *Brassica rapa;* (**C**,**D**) *Rubus ulmifolius* showing witches’ broom and leaves deformation; (**E**,**F**) witches’ broom, leaves and floral tip deformation in *Rosa* sp.

**Figure 4 pathogens-09-00933-f004:**
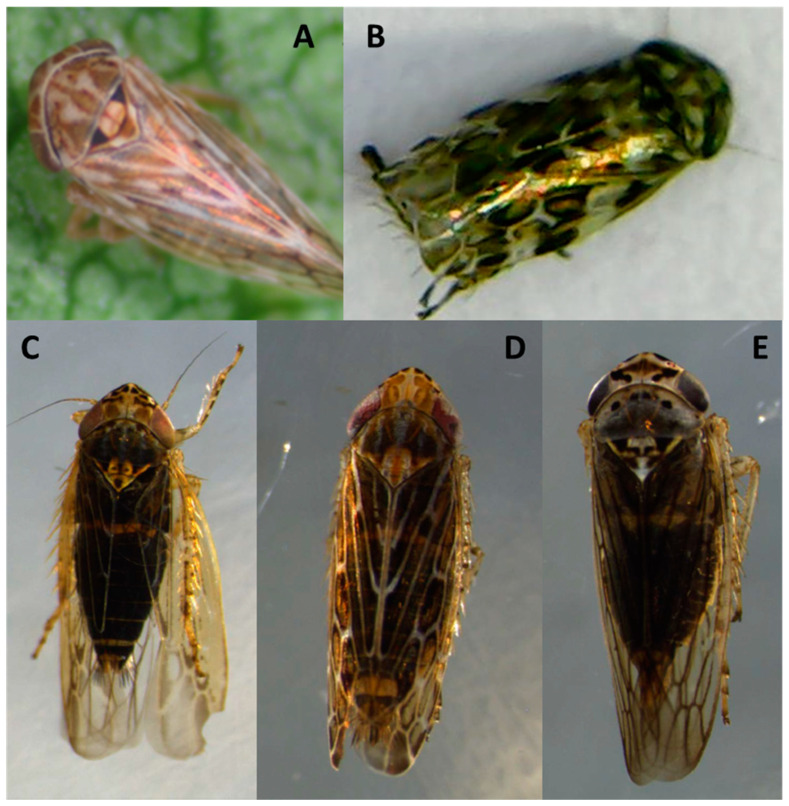
16SrIII-J phytoplasma-positive leafhopper species. (**A**) Bergallia sp.; (**B**) Amplicephalus ornatus; (**C**) Amplicephalus curtulus; (**D**) Amplicephalus pallidus; (**E**) Exitianus obscurinervis.

**Figure 5 pathogens-09-00933-f005:**
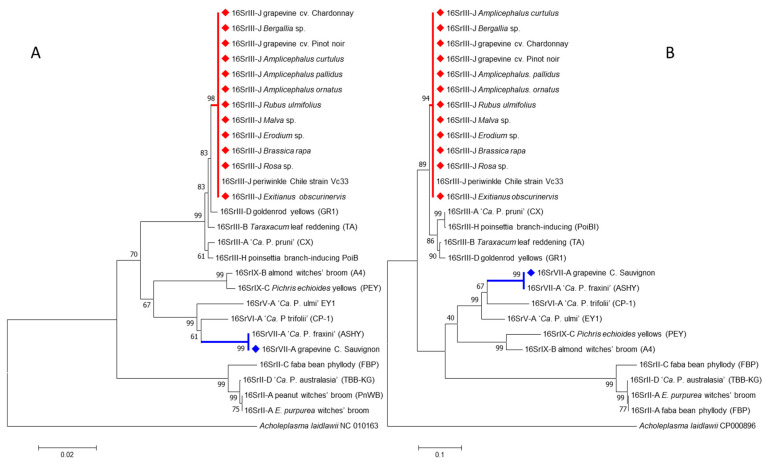
Neighbor-joining analyses of the sequences of 1244 nucleotides obtained from the 16S rRNA gene (**A**) and 710 nucleotides obtained from the *SSu12p* gene (**B**). Samples highlighted with diamonds were obtained in this work. Numbers in nodes represent the bootstrap values obtained from 1000 replicates.

**Figure 6 pathogens-09-00933-f006:**
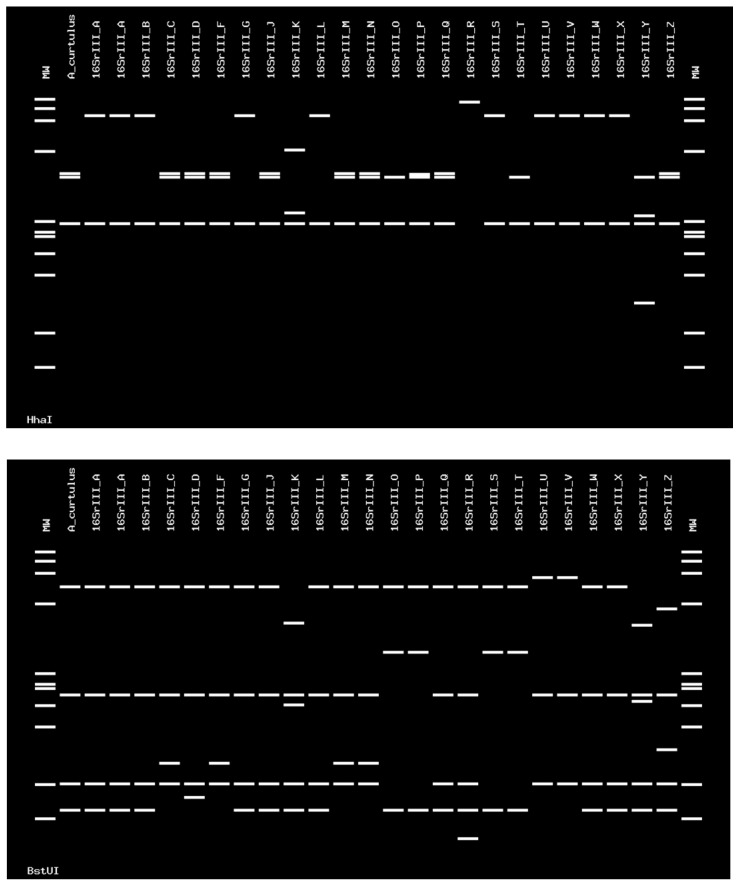
Virtual RFLP profile obtained from the *i*PhyClassifier, using enzymes *Hha*I and *BstU*I. The figure shows identical profiles between the phytoplasma detected in *A. curtulus* and the strain from chayote witches’ broom, classified in the 16SrIII-J subgroup (GenBank accession number AF147706). The phytoplasma detected in the *A. curtulus* showed a similarity coefficient of 1.00 with this strain.

**Figure 7 pathogens-09-00933-f007:**
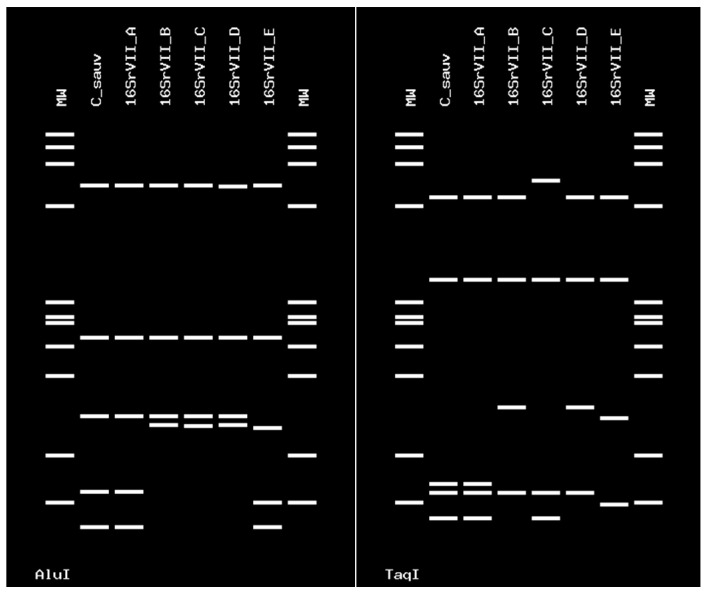
Virtual RFLP profile obtained from the *i*PhyClassifier, using enzymes *Alu*I and *Taq*I. The figure shows identical profiles between the phytoplasmas detected in grapevine cv. Cabernet Sauvignon and the ‘*Candidatus* Phytoplasma fraxini’ 16SrVII-A subgroup (GenBank accession number AF092209). The strain has a similarity coefficient of 1.00 with ‘*Ca*. P. fraxini’.

**Table 1 pathogens-09-00933-t001:** Presence of phytoplasmas in grapevine plants.

Region	Grapevine Samples	Phytoplasma Detected (Number of Positive Samples)
Valparaíso	73	16SrIII-J (17)
O’Higgins	70	16SrIII-J (12), 16SrVII-A (5)
Maule	187	16SrIII-J (26), 16SrVII-A (3)

**Table 2 pathogens-09-00933-t002:** Non-grapevine plant species and insects phytoplasma-positive per vineyard.

Vineyard	Positive Non-Grapevine Plant Species *	Insect	Number of Captured Individual Insects/Analyzed/Positives **
**Casablanca** Valparaíso región	*Brassica rapa* (1) *Malva* spp. (2) *Erodium* spp. (3)	*Paratanus exitiosus*	194/190/120
		*Bergallia valdiviana*	62/60/45
		*Bergallia* sp.	44/40/20
		*Amplicephalus ornatus*	22/20/10
		*Amplicephalus pallidus*	30/30/10
		*Amplicephalus curtulus*	81/80/45
		*Exitianus obscurinervis*	24/20/10
**Marchigue** Libertador O’Higgins region	*Brassica rapa* (1) *Rosa* spp. (7) *Rubus ulmifolius* (3)	*Paratanus exitiosus*	86/85/60
		*Bergallia* sp.	36/35/15
		*Amplicephalus pallidus*	62/60/10
		*Amplicephalus curtulus*	26/25/5
		*Exitianus obscurinervis*	104/100/20
**San Javier** Maule region	*Brassica rapa* (2)	*Paratanus exitiosus*	224/220/160
		*Amplicephalus curtulus*	114/110/20

* ( ) Number of positive plants to 16SrIII-J phytoplasma. ** The insects were analyzed in groups of five individuals.

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
