# Peer review of "Update and New Epidemiological Aspects about Grapevine Yellows in Chile"

_pathogens, 2020, doi:10.3390/pathogens9110933_

Round 1

Reviewer 1 Report

Abstarct:

Line 27: How were the insect species fully or partially classified? Give details in the methods section.

Line 30: What do the Authors refer as spontaneous plants. Are these weeds or alternative hosts of the phytoplasmas?

Line 31: Chile: Authors should think about the applicability of the study globally in other grapevine growing regions. 

Authors should re-write the introduction. This is not in depth at all. Paragraphs should include:

  • Production of grapes in Chile, wine making, tourism and other associated socio-cultural activities.
  • Pests and diseases of grapes in Chile: bacteria, fungi, oomycete, viruses, viroids, then phytoplasmas. Insect pest mealybugs, scale insects, leafhoppers, planthoppers.
  • Economic impact: losses due to damage incured by growers
  • Background: why is the study needed? importance, what information is being sort, what problems are being addressed...
  • Objectives

Materials and methods:

  • How many places were sampled in each region?
  • What was the approximate disease incidence/severity?
  • What method was used for phylogenetic analysis? 
  • Authors should include how many plants of each cultivar were sampled?

Results:

  • Authors should arrange figures as they are discussed in the text.
  • Authors should include virtual gel image of RFLP results.
  • Authors should also include the family of the weed hosts in addition to genuc and species.

The data is promising. Authors should overhaul the manuscript and bring it up to the standard of the publication.

Reviewer 2 Report

Manuscript pathogens-962879 describes a survey of vineyards in Chile for phytoplasmas, alternative hosts and potential insect vectors.  The research is of interest and the manuscript is overall well written.  However, grammatical and typographical errors should be corrected.  See recommendations below.

Specific comments:

Line 23: ... respectively.  Vineyards in ...

Line 25: ... host plants and insects of potential epidemiological relevance.  Five previously ...

Line 26: Change resulted to tested

Line 29: ... potential vectors of 16SrIII-J phytoplasmas.  The ...

Line 36: Viticulture is one of the most important agricultural activities of Chile, and the ...

Line 49: ... as vector of the 16SrIII-J phytoplasma [9].

Lines 50-52: ... presence in Chile, a new survey was conducted, indicating the presence of new potential insect vector, as well as new reservoir plant species in infected vineyards.

Line 54: Change Phytoplasmas to Phytoplasma

Line 56: The sampling of shoots and leaves was performed from 330 grapevine plants and 80 different plant species ...

Line 59: ... were obtained by PCR and nested PCR from ...

Line 60: ... 63 grapevines and 19 non-grape plant species collected ...

Line 61: ... were used to perform phylogenetic ...

Line 66: Change varieties to cultivars

Line 69: Change variety to cultivar

Lines 70-71: ... different symptoms that were positive for the 16SrIII-J phytoplasma were ...

Line 75: ... in R. ulmifolias, a decrease ...

Line 87: Table 1. Presence of phytoplasmas in grapevine plants

Line 97: The 16SrVII-A phytoplasma was not detected in any of the insects (Table 2).

Lines 113-116: Sequences of the 16SrIII-J phytoplasma were 100% identical at the nucleotide level.  These sequences were identical to strain Vc33 from periwinckle in Chile (GenBank accession number LLK01000000).  The 16SrIII-J sequences formed a monophyletic group, as sown by neighbor joining analyses (Figure 1).  In addition, the sequence ...

Line 120: ASHY strain infecting periwinckle at the University of Bologna ...

Lines 126, 127, 128 and 129: Change variety to cultivar

Line 135: Add a space between phytoplasma and presence

Lines 150-151: ... were detected in vineyards of Chile during this survey.  Five wild plant species were positive for phytoplasma ...

Line 154: The 16SrIII-J phytoplasma was prevalent.  This result differs from previous surveys ...

Line 155: Eliminate before

Lines 163-164: ... of P.exitiosus.  This was the most abundant ...

Lines 164-165: ... were positive for the 16SrIII-J phytoplasma.  A special ...

Line 165: Change variety to cultivar

Line 166: Change from to by

Line 167: ... observed on white cultivars.

Line 168: ... fact that the cultivars was introduced to Chile in the 16th ...

Line 177: Eliminate the

Line 180: Change its to their

Line 180: Change stays to stay

Lines 181-182: ... reservoir of the 16SrIII-J phytoplasma during the ...

Line 191: ... with the 16SrIII-J and 16SrVII-A phytoplasmas.  It should be ...

Line 192: Change potentially to potential

Line 202: Each samples corresponded to four 30 cm-long shoots with ...

Line 208: Change whit to with

Line 213: Eliminate they

Line 214: Change morph anatomical to morphological

Lines 216-217: Eliminate During this activity

Line 223: Change 30s to 30S

Line 228: Add a period after (Promega)

Round 2

Reviewer 1 Report

The authors have submitted a manuscript, "Update and new epidemiological aspects about grapevine yellows in Chile" describes the incidence and characterization of grapevine yellows phytoplasma in grapevines in Chile.

- Authors should insert the program used or phylogenetic analysis as claimed. Which evolutionary model was used to model distances.
